# Human Amniotic Mesenchymal Stem Cells and Fibroblasts Accelerate Wound Repair of Cystic Fibrosis Epithelium

**DOI:** 10.3390/life12050756

**Published:** 2022-05-19

**Authors:** Elisa Beccia, Valeria Daniello, Onofrio Laselva, Giorgia Leccese, Michele Mangiacotti, Sante Di Gioia, Gianfranco La Bella, Lorenzo Guerra, Maria Matteo, Antonella Angiolillo, Massimo Conese

**Affiliations:** 1Department of Clinical and Experimental Medicine, University of Foggia, 71122 Foggia, Italy; elisa.beccia@unifg.it (E.B.); valeria.daniello@unifg.it (V.D.); onofrio.laselva@unifg.it (O.L.); giorgia_leccese.555553@unifg.it (G.L.); sante.digioia@unifg.it (S.D.G.); 2Department of Medicine and Health Sciences “V. Tiberio”, University of Molise, 86100 Campobasso, Italy; 3S.C. di Radioterapia Oncologica, Policlinico Riuniti, Azienda Ospedaliero-Universitaria, 71122 Foggia, Italy; michele.mangiacotti@unifg.it; 4Istituto Zooprofilattico Sperimentale della Puglia e della Basilicata (IZSPB), 71121 Foggia, Italy; gianfranco.labella@izspb.it; 5Department of Biosciences, Biotechnologies and Biopharmaceutics, University of Bari, 70125 Bari, Italy; lorenzo.guerra1@uniba.it; 6Department of Medical and Surgical Sciences, University of Foggia, 71122 Foggia, Italy; maria.matteo@unifg.it

**Keywords:** human mesenchymal stem cells, fibroblasts, cystic fibrosis, airway epithelium, wound repair, cell proliferation

## Abstract

Cystic fibrosis (CF) airways are affected by a deranged repair of the damaged epithelium resulting in altered regeneration and differentiation. Previously, we showed that human amniotic mesenchymal stem cells (hAMSCs) corrected base defects of CF airway epithelial cells via connexin (CX)43-intercellular gap junction formation. In this scenario, it is unknown whether hAMSCs, or fibroblasts sharing some common characteristics with MSCs, can operate a faster repair of a damaged airway epithelium. A tip-based scratch assay was employed to study wound repair in monolayers of CFBE14o- cells (CFBE, homozygous for the *F508del* mutation). hAMSCs were either co-cultured with CFBE cells before the wound or added to the wounded monolayers. NIH-3T3 fibroblasts (CX43+) were added to wounded cells. HeLa cells (CX43-) were used as controls. γ-irradiation was optimized to block CFBE cell proliferation. A specific siRNA was employed to downregulate CX43 expression in CFBE cells. CFBE cells showed a delayed repair as compared with wt-CFTR cells (16HBE41o-). hAMSCs enhanced the wound repair rate of wounded CFBE cell monolayers, especially when added post wounding. hAMSCs and NIH-3T3 fibroblasts, but not HeLa cells, increased wound closure of irradiated CFBE monolayers. CX43 downregulation accelerated CFBE wound repair rate without affecting cell proliferation. We conclude that hAMSCs and fibroblasts enhance the repair of a wounded CF airway epithelium, likely through a CX43-mediated mechanism mainly involving cell migration.

## 1. Introduction

Cystic fibrosis (CF) is due to mutations occurring in the CF Transmembrane Conductance Regulator (*CFTR*) gene, leading to the lack/dysfunction of the CFTR protein at the apical side of absorptive/secretive epithelia [1]. A pathogenic cascade, stemming from CFTR protein abnormalities, leads to epithelial sodium channel (ENaC) hyperactivity, heightened resorption of sodium and water from the periciliary fluid [2], mucociliary clearance annihilation and eventually to opportunistic bacterial infections, inflammation and disruption of bronchiolar structure due to abnormal repair of the airway epithelium [3]. CF lung pathology is represented in the overt disease by many alterations at the bronchial mucosa level, such as hyperplasia of goblet and basal cells [4,5,6,7], squamous metaplasia [6,8], increase in epithelium height [5,7,9], cell shedding [4,5,6,9,10] and subepithelial fibrosis [4,11]. The mechanism by which these alterations occur is not completely clear. However, it is thought that derangement in epithelium repair, regeneration and remodeling is involved in the pathogenesis of CF lung disease and that CF Transmembrane Conductance Regulator (CFTR) lack/dysfunction plays a role in the dysregulation of the steps leading to a correct repair [12]. CFTR lack/dysfunction has been recently linked to epithelial–mesenchymal transition (EMT), a fundamental mechanism in epithelial repair [12]. Moreover, both pro-inflammatory mediators [7] and bacterial products [13,14,15,16] may cause an injury on the airway epithelium; thus, it is not easy to understand this complex interplay and which mediator is playing a major role. The discovery of the exact mechanism leading to wound healing would accomplish the fruitful result of bringing to the appraisal of biomarkers of this complex process. In general, some biomarkers of wound healing have been already indicated, such as cytokines, growth factors and proteases [17,18]. The knowledge of specific biomarkers of wound healing in CF would assist in those individuals that have an impaired ability to heal.

The repair of a CF epithelium has been mainly studied in in vitro models by procuring an injury by scratching with a pipette tip [15,19,20,21,22,23,24,25], using lethal electroporation [26,27] or by an airbrush linked to a pressure regulator [28,29]. All these studies, performed on either immortalized or primary cell lines, have unequivocally shown that CF cells, compared to non-CF cells, display a defect in recovering the wound and that lower cell migration and/or proliferation is mostly responsible for this phenotype. However, few studies have focused on the molecular mechanism that could reveal biomarkers of the dysregulated wound repair of the CF epithelium, including those transcription factors involved in EMT, such as TWIST1 [24], YAP1 [30] and KLF4 [25], and the amino acid transporter SLC6A14 [31].

Human amniotic mesenchymal stem cells (hAMSCs) are a source of cell-based therapies due to their interesting features: they are ethically acceptable due to their derivation from the placenta, which is discarded at parturition, and they express staminal markers and can differentiate into epithelial-like cells acquiring CFTR expression [32]. Moreover, hAMSCs can also be considered in the context of allogeneic therapies due to the lack of MHC antigens and of co-stimulating activities. We recently demonstrated the cellular therapy potential of hAMSCs for CF lung disease using co-cultures of hAMSCs with CFBE41o- cells [32]. In this model, hAMSCs acquired an epithelial-like phenotype and CFTR expression in their plasma membrane. Moreover, we have shown in CFBE–hAMSCs co-cultures the recovery of some of the basic defects presented by CFBE cells, namely absent CFTR-mediated chloride channel activity, epithelial sodium channel (ENaC)-mediated fluid hyperabsorption and alterations in actin cytoskeletal and tight junction organization and function [33]. More recently, we demonstrated that the rescue of CFTR protein expression and function, and airway epithelium tightness, by hAMSCs is due to gap junction (GJ) intercellular communication [34].

Fibroblasts are stromal cells that provide the majority of the structural framework of almost all types of tissues, i.e., the stroma [35]. Since their main role is the secretion of extracellular matrix molecules, mainly collagen, and as the different types of collagen are the major component of tissues such as bone, cartilage and skin, fibroblasts also have significant roles in tissue development, maintenance and repair [36]. Due to these characteristics, they have been used for clinical studies mostly for wound healing treatments [37,38].

The anti-inflammatory and antibacterial properties of MSCs in vitro and in vivo CF murine models have been investigated [39,40], while the role of MSCs in general, and of hAMSCs in particular, in the wound repair of a damaged airway epithelium has not been studied yet. Herein, we have explored the capacity of hAMSCs and fibroblasts in the wound repair exerted by a mechanical injury of the CF airway epithelium, taking CFBE cells as such a model. hAMSCs were used either in co-culture or added to a wounded CFBE epithelium, demonstrating that hAMSCs accelerated wound repair. Fibroblasts displayed the same capacity as well when added to wounded CFBE cells.

## 2. Materials and Methods

### 2.1. hAMSC Cultures and Phenotype

hAMSCs were isolated from the amniotic membrane of term placenta (*n* = 5, all women <40 year of age) and grown in advanced DMEM supplemented with 10% FBS, 55 μM β-mercaptoethanol, 1% L-glutamine, 1% penicillin/streptomycin and 10 ng/mL epidermal growth factor (EGF) (Sigma-Aldrich, Milan, Italy) [33]. They were subcultured until passage 5, when 80% of confluence was obtained. For phenotypic analysis, cells were then stained with fluorochrome-conjugated monoclonal antibodies against hematopoietic (CD14, CD34, CD45) or mesenchymal (CD29, CD73, CD105) markers, as previously published [41]. All antibodies were purchased from Invitrogen, Thermo Fisher Scientific (Waltham, MA, USA), and were conjugated with fluorescein isothiocyanate (FITC), except the antibody against CD73 that required an additive incubation with secondary antibody (FITC goat anti-mouse; Sigma-Aldrich, Milan, Italy) for 30 min at 4 °C. At least 1 × 10^6^ cells were finally analyzed using the Amnis Flowsight IS100 (Merck KGaA, Darmstadt, Germany). Analysis of positive cells was obtained with the IDEAS 6.0 Software (Amnis, EMD Millipore, Seattle, WA, USA).

The usage of hAMSCs was conducted according to the guidelines of the Declaration of Helsinki and approved by the Ethics Committee of the Ospedali Riuniti of Foggia (n. 138/CE/2020 issued on 30 November 2020). Informed consent was obtained from all women included in the study.

### 2.2. Cell Cultures

Human immortalized bronchial epithelial cell lines were 16HBE14o- (16HBE), expressing wild-type CFTR, and CFBE41o- (CFBE), homozygous for the *F508del* allele [42]. Cells were grown at 37 °C under 5% CO_2_ on flasks in MEM medium (Costar, Corning, MA, USA) containing 10% FBS, 1% L-glutamine and 1% penicillin/streptomycin (all purchased from Thermo Fisher Scientific, Milan, Italy). The NIH mouse fibroblast cell line (NIH-3T3 L1, a kind gift of Dr. Giuseppe Procino, University of Bari “A. Moro”, Italy) was grown in Dulbecco’s modified eagle medium (DMEM) with 4.5 g/L glucose and sodium pyruvate (Corning), containing 10% FBS, 1% L-glutamine and 1% penicillin/streptomycin. HeLa cells [43] were grown in the same medium as NIH-3T3 cells.

### 2.3. Co-Cultures onto 24-Well Plates

Co-cultures were obtained by mixing hAMSCs, at passages 2–5, with CFBE cells at 1:5 ratio (1.5 × 10^4^ hAMSCs with 6 × 10^4^ CFBE cells). Mixed cells were seeded onto 24-well plates and allowed to become confluent after 6 days of culture.

### 2.4. Wound Repair Assay in 24-Well Plates

Airway epithelial cells were injured mechanically with a P10 pipette tip [21]. A mark on the 24 well allowed us to photograph the wounds at exactly the same place at various times (time 0 and after 6 h, 24 h or 48 h). Digital images of 10× fields were obtained with a Leica DM IRB inverted microscope equipped with a Leica DFC450 C camera (Leica, Wetzlar, Germany). The percentage and the rate of wound closure, presented in μm/h, were calculated with ImageJ software (National Institutes of Health, Bethesda, MD, USA) from the wound size measured after repair compared with the initial wound area. At each time point, images were converted to 8-bit (menu Image > Type > 8-bit), contrast was increased (menu Process > Enhance Contrast) and then binary (black and white) images were created, white corresponding to the cell sheet (menu Image > Adjust > Threshold). The wound-closure rate was calculated by dividing the difference in wound closure between two time points by the time interval and was expressed as μm/h.

The scratch assay was carried out with monolayers of 16HBE, CFBE, hAMSC–CFBE co-culture and after adding hAMSCs to a wounded CFBE monolayer. To know the number of hAMSCs to be added at a 1:5 ratio, CFBE cells were detached and counted (TC^10^ Automated Cell Counter, Bio-Rad, Singapore) after wounding.

To analyze the effect of CX43 siRNA on wound closure, CFBE cells were irradiated (see Section 2.6) or not and then plated. After an overnight incubation, cells were at 70% of confluency and were transfected with active siRNA pool directed against Cx43 (30 nM; Riboxx GmbH, Dresden, Germany) as we previously carried out [34]. After 24 h, cells were wounded and evaluated for the percentage of wound closure at t0, and after 24 h and 48 h. No toxic effects determined by the transfection were observed. In the same experiments, untransfected and cultures transfected with scrambled negative control siRNA (Riboxx) were included. The percentage of wound closure was evaluated as described above.

### 2.5. CM-DiI Labelling of hAMSCs

hAMSCs were labelled with chloromethylbenzamido (CellTracker CM-DiI, Thermo Fisher Scientific, Waltham, MA, USA), as previously performed [32], and co-cultured with CFBE cells for 6 days or added to wounded CFBE monolayers. At 0, 6 h, 24 h and 48 h post wounding, monolayers were observed at Zoe Fluorescent Cell Imager (Bio-Rad, Segrate, Italy) in the red channel (ex. 615–661 nm, em. 556–620 nm). Monolayers did not show any autofluorescence.

### 2.6. Irradiation of CFBE Cells

To block cell proliferation, CFBE cells (7.5 × 10^4^) were irradiated with a ^137^Cs gamma irradiator IBL-437 (CIS Bio International, Schering SA, Gif Sur Yvette, France) at doses of 10, 15 and 20 Gy. After irradiation, cells were plated onto 24-well plates for 0, 24 h and 48 h and evaluated for cell proliferation by different methods (see Section 2.7).

### 2.7. Cell Proliferation Assays

Cells were either trypsinized and counted in a Bürker chamber or stained with DAPI (4′,6-diamidino-2-phenylindole, Sigma-Aldrich) while still attached to the plate by the following protocol: cells were fixed with freshly made 4% paraformaldehyde and 2% sucrose dissolved in PBS (pH 7.4) for 5 min at room temperature, washed three times with PBS, permeabilized with 20 mM HEPES, 20 mM sucrose, 20 mM NaCl, 20 mM MgCl_2_, 0.5% Triton X-100, 0.5% triton X-10, washed three times with PBS and added with 0.05% *v*/*v* DAPI for 5 min at room temperature. Finally, cells were analyzed at the Zoe Fluorescent Cell Imager in the blue channel (ex. 358 nm, em. 461 nm). For each condition, three random 20x fields were chosen to count stained nuclei. The same digital images were analyzed for fluorescence (mean) by the Image J software.

To analyze the effect of CX43 siRNA on cell proliferation, CFBE cells were either irradiated or not, plated at 1 × 10^4^ in a 96-well plate and transfected with either active siRNA or scrambled siRNA. After 24 h, cell proliferation was evaluated at t0 and after 24 h and 48 h by the MTT (3-(4,5-dimethylthiazol-2-yl)-2,5 diphenyl tetrazolium bromide) assay, as previously described [44]. The cell viability, considered as a proxy to cell number, was calculated as follows:

% viability = [(Optical density {OD} of treated cell − OD of blank)/(OD of vehicle control − OD of blank) × 100], considering untransfected cells at t0 as 100%.

### 2.8. Wound Repair on Irradiated Monolayers (60 mm Dishes)

CFBE cells were seeded onto 60 mm Petri dishes (1 × 10^6^). Once confluent monolayers were wounded, subjected to irradiation (15 or 20 Gy) and analyzed for wound closure. To know the number of hAMSCs/HeLa cells/NIH-3T3 cells to be added in a 1:5 ratio, CFBE cells were detached and counted (TC^10^ Automated Cell Counter, Bio-Rad, Singapore) after irradiation and wounding. In parallel, 16HBE cells were evaluated. The percentage and rate of wound closure were evaluated as for the 24-well experiments.

### 2.9. Statistical Analysis

The statistical analysis was carried out using Prism for Windows, version 5.01, GraphPad Software Inc., San Diego, CA, USA. The ANOVA with Tukey’s post hoc test was used. Differences were considered significant when *p* < 0.05.

## 3. Results

### 3.1. Phenotypic Characteristics of hAMSCs Are Maintained until p5

hAMSCs possessed a spindle shape, elongated morphology and adherence to plastic and could be kept in culture until passage 5 (p5) without any change in morphology. By cytofluorimetry, they expressed MSC markers, such as CD29, CD105 and CD73, but showed negligible expression of hematopoietic markers (CD14, CD34, CD45: all by 3–4% of cells). Marker expression was maintained from p2 until p5 (Appendix A). Based on these results, we proceeded with further experiments using p2-p5 hAMSCs.

### 3.2. Wound Closure Is Delayed in CFBE Cells

CFBE cells closed the wound only after 48 h (Figure 1A(a–c)), while non-CF 16HBE cells closed the wound in 24 h (Figure 1A(d,e)). Both the % of wound size in comparison with the initial area (0.1 ± 1.9 vs. 28.6 ± 9.1 (mean ± SD); *p* < 0.0001) and the wound-healing rate (1.8 ± 0.3 vs. 0.7 ± 0.2 μm/h; *p* < 0.0001) at 24 h were significantly different between 16HBE and CFBE (Figure 1B,C).

### 3.3. hAMSCs Accelerate Wound Closure of CF Monolayers

To understand whether hAMSCs could have a role in wound closure of a CF monolayer, we either wounded co-cultures (hAMSC–CFBE 1:5 ratio) or applied hAMSCs to a wounded CF monolayer. In this case, the wound was also studied at a shorter time (6 h) to investigate if hAMSCs were capable of closing it in a faster way. Results showed that, in both experimental settings, the presence of hAMSCs accelerated the wound closure at 6 h and 24 h (Figure 2A; see for comparison panels b, f, j for 6 h and c, g, k for 24 h). Interestingly, CFBE cells migrated into the wound, maintaining their cobblestone morphology (Figure 2A(b)), whereas in the co-culture it was possible to see both epithelial and fibroblast-like cells in the middle of the wound (Figure 2A(k)). When hAMSCs were added to the wounded CF monolayers, the wound presented predominantly spindle-like cells closing it (Figure 2A(g)). Interestingly, the closure at 24 h was complete when adding hAMSCs, whereas at the same time point with co-cultures the wound was not completely closed (Figure 2A(g,k)).

The evaluation of wound size (in % of the control, i.e., time 0) confirmed that adding hAMSCs to the injured CFBE closed the wound already at 24 h, whereas with co-cultures the wound size decreased by 80% (Figure 2B). In the interval of 0–6 h, the wound closure rate was higher with hAMSCs added after the wound and with co-cultures as compared with CFBE only (Figure 2C). On the other hand, the rate declined with all conditions but remained higher when hAMSCs were added as compared with CFBE only, with a significant difference both at 6–24 h and 0–24 h.

To further confirm that hAMSCs were migrating/proliferating and accelerating CFBE monolayer wound closure, they were labelled with the vital dye CM-Dil and added to injured CFBE. As shown in Appendix A, CFBE monolayers did not display any autofluorescence, indicating that the signal associated with hAMSCs was specific. When added to injured CFBE monolayers, fluorescent hAMSCs were found to distribute over CF cells and in the middle of the wound (Appendix A). As early as 6 h, hAMSCs increased in their number in the middle of the wound (Appendix A), while at 24 h the wound was filled by both hAMSCs and CFBE cells (Appendix A). In the co-cultures, there were very few fluorescent hAMSCs at 6 h between flaps (Appendix A), while they increased at 24 h (Suppl. Appendix A). As with non-fluorescent hAMSCs, the co-culture did not completely repair the wound at 24 h (Appendix A). Overall, these data confirmed that the condition CFBE + hAMSCs repairs faster than co-cultures and hAMSCs participate in wound healing.

### 3.4. Block of Cell Proliferation Retards CFBE Wound Closure

To establish the relative role of cell proliferation and migration in the wound closure, CFBE cells were γ-irradiated to block proliferation irreversibly. While 10 Gy radiation exposure did not affect cell proliferation (as cell number increased at 24 h and 48 h, such as in the non-irradiated controls), 15 and 20 Gy radiation blocked the increase in cell numbers, and with 20 Gy we even observed a slight reduction compared to controls (Appendix A). These results were confirmed by the staining of nuclei with DAPI and evaluation of their counts (Appendix A) and their associated fluorescence (Appendix A). Based on these results, the following experiments were carried out by treating CFBE with 15 Gy.

To confirm which γ-ray dose was helpful in studying wound closure, CFBE cells were irradiated and evaluated in the scratch assay onto 60 mm dishes (Appendix A). Irradiation with 15 Gy retarded the wound closure, whereas considerable sloughing of cells off the monostrate was observed with 20 Gy, mostly at 48 h (Appendix A), as confirmed by the confluency evaluation (Appendix A). All in all, these data determined the 15 Gy-dose be useful for further experiments with hAMSCs.

Figure 3 shows that, as with 24-well plates, injured CFBE monolayers were delayed in wound closure compared to 16HBE in 60 mm dishes (d–f vs. a–c). Irradiated CFBE cells showed a further retardation of wound closure, suggesting that cell proliferation may play a role (Figure 3A(g–i)). However, when hAMSCs were added to wounded CFBE, the closure accelerated at 24 h, but the wound was completely closed only at 48 h (Figure 3A(j–l)). Figure 3B reports the wound size under these experimental conditions. While with 16HBE the wound size reduced to zero at 24 h, CFBE cells showed a higher wound size than 16HBE at 24 h, a defect further aggravated by irradiation. However, in the presence of hAMSCs the wound size reduced to zero only at 48 h, indicating that hAMSCs were delayed in their wound healing properties in the presence of CFBE blocked in their proliferation. Figure 3C shows that hAMSCs recovered the wound closure rate of irradiated CFBE, although not at the levels of 16HBE.

Overall, these results indicate that CFBE proliferation contributes to wound closure and confirm that hAMSCs play a significant role in accelerating wound closure of a damaged CFBE monolayer.

### 3.5. NIH-3T3 Cells, but Not HeLa Cells, Accelerate Wound Closure

In order to understand whether CX43 plays a role in wound repair of a wounded and irradiated CFBE monolayer, we used, instead of hAMSCs, NIH-3T3 fibroblasts, positive for CX43 [45], and HeLa cells, negative for CX43 expression [46,47]. Results showed that NIH-3T3 cells accelerated the wound repair of irradiated wounded CFBE monolayers even better than hAMSCs, since the wound was closed already at 24 h (Figure 4A,B), while the wound remained open in the presence of hAMSCs for around 20% as compared with time 0 (compare with Figure 3B). Moreover, the wound closure rate was higher in the presence of NIH-3T3 cells at 0–24 h as compared with CFBE (Figure 4C) and with hAMSCs (compare with Figure 3C).

On the other hand, HeLa cells did not have this property. Indeed, the addition of HeLa cells did not close the wound neither at 24 h nor at 48 h (Figure 5A). However, the wound size decreased significantly more in the presence of HeLa cells at both time points (Figure 5B), while the wound closure rate was not different in both conditions at all the time intervals (Figure 5C).

The fact that the addition of NIH-3T3 cells, but not Hela cells, operated a faster wound closure than CFBE only strongly indicates that there could be a role for CX43 in wound repair.

### 3.6. CX43 siRNA Accelerates CFBE Wound Closure without Affecting Cell Proliferation

To further explore the role of CX43 in proliferation and wound healing, downregulation of CX43 was investigated by a specific siRNA. We have already shown that basal CX43 mRNA levels were comparable between the CFBE cells and hAMSCs, although non-statistically higher levels were found in hAMSCs. As to the protein, a specific signal in CFBE cells was localized at the cell borders while hAMSCs were characterized by Cx43 localization both at the cell membrane and within the cytoplasm. In addition, we could successfully downregulate CX43 mRNA (by ~40%) and protein levels in CFBE cells by a specific siRNA against CX43 [34]. CFBE cells were either irradiated or not, transfected with active or scrambled siRNAs as control and evaluated for cell proliferation by the MTT assay. Non-irradiated cells proliferated, whereas irradiated cells were slower in proliferation. Cell proliferation of non-irradiated and irradiated CFBE cells was not affected by CX43 siRNA transfection (Figure 6A,B). SiRNA targeted to Cx43 significantly enhanced the rate of wound closure in both irradiated and non-irradiated CFBE monolayers at 24 h and 48 h (Figure 6C,D), whereas the transfection of scrambled siRNA (siRNA-) was with no effect. In non-irradiated cells, the increase in wound closure upon active siRNA transfection was cumulatively 29–57% as compared with untransfected cells, whereas in irradiated cells this increase was 25–27%.

## 4. Discussion

In CF, where more than 2000 sequence variations are ascribed to six mutation classes, a cell-based therapy may represent an agnostic source of cell-based treatments, that, independently on the mutation class, may be useful to obtain anti-bacterial, anti-inflammatory, immunomodulatory and pro-regenerative effects [48]. Although many studies have evidenced the anti-inflammatory and anti-microbial effects of MSCs in various in vitro and in vivo CF models [40,49,50], no reports have shown their pro-resolutive actions in the wound repair of a CF airway epithelium. This consideration holds also for fibroblasts.

In this paper, for the first time, we present in vitro data showing that MSCs obtained from an ethical source, i.e., the amniotic membrane, can accelerate the defective wound closure presented by CFBE monolayers. These data complement those previously obtained by our group showing acquisition of CFTR expression by hAMSCs in co-culture with CFBE cells [32] and therapeutic properties of hAMSCs towards the CF-associated basic defects, such as reduced chloride efflux, fluid hyperabsorption and lack of epithelial tightness [33,34].

The wound repair at the epithelial level is a combination of cell migration and cell proliferation. In vivo, in the lungs, there is a chain of events contemplating spreading and migration of neighboring epithelial cells, a further step characterized by migration and proliferation of progenitor cells and finally differentiation [51]. Studies on airway human xenografts in immunodeficient mice have further defined that the main steps in airway epithelial repair and reconstitution are cell migration (step I), proliferation (step II), epithelial pseudostratification (step III) and mucociliary epithelial differentiation (step IV) [52]. These processes were deranged in the regeneration of a CF airway epithelium, and the heightened proliferation was suggested to be involved in the delay observed in CF epithelial differentiation [53]. Other studies have presented contrasting results about proliferation and migration of CF airway epithelial cells, depending on the considered cell types [19,20,21,22,24,26]. In our study, we blocked CFBE proliferation by γ-irradiation and found that the wound closure was retarded, suggesting that cell proliferation contributes to wound repair. However, even in this condition hAMSCs accelerated the wound closure, showing that the amount of hAMSCs added to the wound was sufficient to repair a defective CF epithelial monolayer.

By allowing the formation of GJ intercellular communication (GJIC), CXs permit the passage of ions and small solutes and are involved in the regulation of proliferation/differentiation and the maintenance of tissue homeostasis [54,55]. GJs have been shown to be involved in the wound healing of skin and cornea [56], with a complex expression pattern of CXs at various stages of wound repair [57]. However, scant information is available for the repair of a wounded airway epithelium [58,59]. Given the recognized importance of epithelial junctions in the pathophysiology of CF [60], this gap should be filled. We have previously shown that hAMSCs and CFBE can form CX43-mediated GJs and that this coupling is involved in the rescue of CFTR function and recovery of other defects presented by CFBE cells. In order to see whether CX43-GJIC played a role in the wound repair, we used two other cell lines that presented (NIH-3T3) or not (HeLa) CX43 expression. Based on the results presented herein, i.e., NIH-3T3 cells, but not HeLa cells, induced a faster wound closure when added to the damaged CFBE monolayer, we infer that GJIC plays a role in the wound healing process. Indeed, downregulation of CX43 in CFBE cells accelerated the wound closure rate, while not affecting cell proliferation, and these results are in accordance with those found with a CX43 siRNA in human keratinocytes [61]. To reconcile results obtained in CX43 siRNA-treated CFBE cells with those achieved in NIH-3T3 fibroblasts, it should be recalled that CX43 is downregulated during epidermal wound repair [56,62,63], which appears to be a prerequisite for the coordinated proliferation and mobilization of keratinocytes during wound healing. Interestingly, non-expressing CXs HeLa cells did not accelerate CFBE wound closure. These results can be explained by the fact that HeLa cells are endowed with an inherent defect in the wound repair, likely for a reduced capacity in cell motility [64]. Overall, based on these results, we suggest that either spontaneously (hAMSCs, NIH-3T3) or instigated by siRNA (CFBE), healing cells downregulate CX43 and allow better cell migration ensuing in a more rapid wound closure.

Finally, we show herein that NIH-3T3 fibroblasts are also capable of closing the wound of a damaged CF epithelium at a faster rate of hAMSCs. This is not surprising, since fibroblasts show a lot of similarities to MSC, including proliferation, differentiation, surface marker expression and immunosuppression [65]. Due to these characteristics, they have been used for clinical studies mostly for wound healing treatments [37,38]. However, others have found that colony-forming capacity and differentiation potential are specific important properties that discriminate MSCs from fibroblasts [66]. On the other hand, homing and migration are unique to MSCs [67], and this property is very interesting when we should think to the administration of hAMSCs to CF patients, ideally reaching the progenitor/stem cell niche identified at the levels of basal layer of the airway epithelium [68]. Finally, while MSCs are characterized by exhaustion of their regenerative and immunosuppressive capabilities [69,70], fibroblast might be induced to hyper-proliferate and produce fibrosis [38]. Besides in vitro studies, such as ours, further exploration of whether cell-based therapies with hAMSCs/fibroblasts are feasible and translatable would benefit from in vivo CF animal studies with relevant infection-driven epithelial injury [13].

While biomarkers of inflammation in CF have been found and validated [71], specific molecular biomarkers of the wound healing process of CF airways have not been thoroughly considered, with some studies providing a limited list of them. Our data suggest that CX43 expression dysregulation may represent such a biomarker.

In summary, hAMSCs and fibroblasts accelerated the wound closure of scratched and low-proliferating CFBE cells, and CX43 downregulation was implied in this effect, indicating that further analysis on CX43 functions in a damaged CF airway epithelium is warranted.

## 5. Conclusions

In this paper, we have demonstrated that MSCs obtained from the amniotic membrane and murine fibroblasts can give rise to a faster repair of a mechanically damaged CF airway epithelium. Our data strongly suggest that CX43 downregulation might be responsible for some events actuating wound closure, above all cell migration. Future studies will be devoted to understanding the molecular cues intertwined with CX expression and function during wound healing. Such findings suggest that MSC treatment may be a potential emerging intervention as a cellular option for CF patients.

## Figures and Tables

**Figure 1 life-12-00756-f001:**
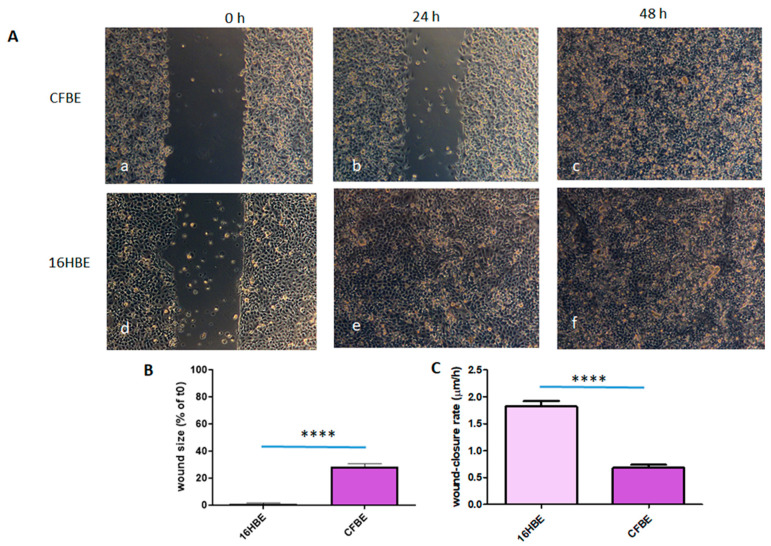
Wound closure in CFBE vs. 16HBE cells. (**A**) Images of CFBE (a–c) and 16HBE (d–f) at 0 (a,d), 24 h (b,e) and 48 h (c,f) after the wound. Bar = 100 μm. (**B**) Percentage of wound size after 24 h. Data are represented as a percentage of the initial area of the wound (t0) considered as 100%. **** *p* < 0.0001. (**C**) Wound-closure rate (μm/h) over a 24 h period. **** *p* < 0.0001. Results are mean ± SD of three experiments.

**Figure 2 life-12-00756-f002:**
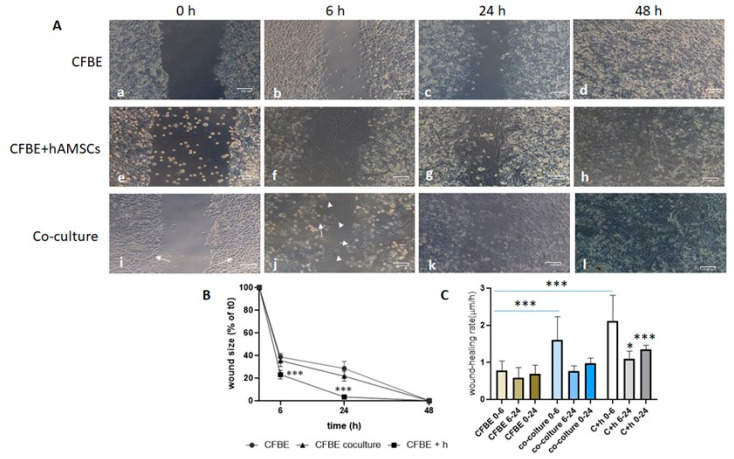
Wound closure in the presence of hAMSCs. (**A**) Injury was determined on CFBE (a–d), CFBE with the addition of hAMSCs (6 × 10^4^) after wound (e–h) and hAMSC–CFBE co-cultures (i–l). Wound closure was evaluated at 0 h (a,e,i), 6 h (b,f,j), 24 h (c,g,k) and 48 h (d,h,l). White arrows in i and j denote hAMSCs, while arrow heads in j point to cobblestone epithelial cells. Bar = 100 μm. (**B**) Percentage of wound size after 6, 24 and 48 h. Data are represented as a percentage of the initial area of the wound (t0) considered as 100%. *** *p* < 0.001 CFBE + h vs. CFBE at 6 and 24 h. (**C**) Comparison of wound closure rates post wounding. * *p* < 0.05 C + h 6-24 vs. CFBE 6-24; *** *p* < 0.001 C + h 0–24 vs. CFBE 0–24. Data were collected from five experiments and are shown as mean ± SD.

**Figure 3 life-12-00756-f003:**
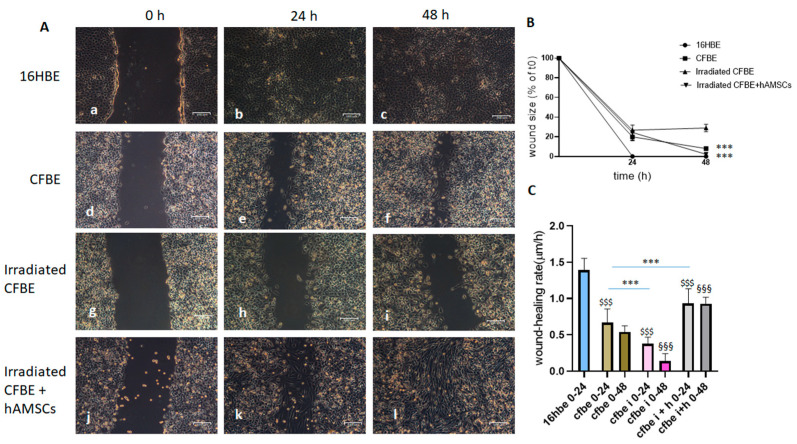
Wound closure of irradiated monolayers in the presence of hAMSCs. (**A**) Injury was determined on 16HBE (a–c), CFBE (d–f), irradiated CFBE (g–i) and irradiated CFBE with the addition of hAMSCs (6 × 10^5^) after wounding (j–l). Wound closure was evaluated at 0 h (a,d,j,g), 24 h (b,e,h,k) and 48 h (c,f,i,l). Bar = 100 μm. (**B**) Percentage of wound size after 24 and 48 h. Data are represented as a percentage of the initial area of the wound (t0) considered as 100%. *** *p* < 0.001 both irradiated CFBE and irradiated CFBE + hAMSC vs. CFBE at 48 h. (**C**) Wound-closure rate (μm/h) over 24 h and 48 h time intervals. *** *p* < 0.001; ^$$$^
*p* < 0.001, 16HBE 0–24 vs. CFBE 0–24, CFBE i 0–24 and CFBE i + h 0–24; ^§§§^
*p* < 0.001, CFBE 0–48 vs. both CFBE i 0–48 and CFBE i + h 0–48. Data were collected from three experiments and are shown as mean ± SD.

**Figure 4 life-12-00756-f004:**
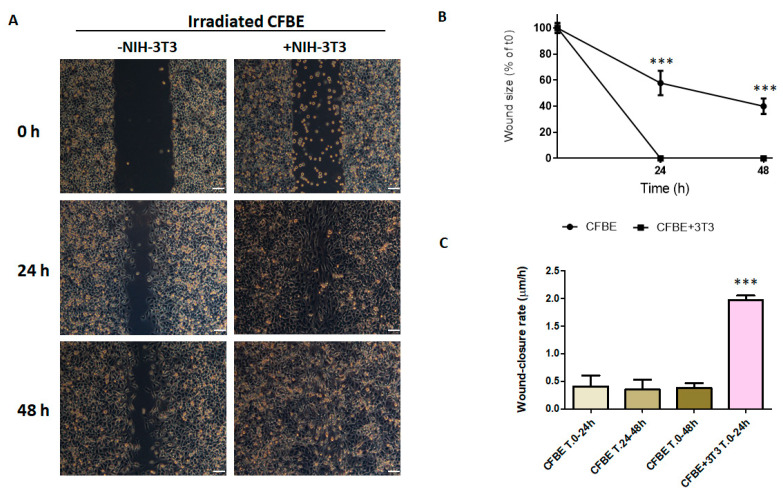
Wound closure of irradiated monolayers in the presence of NIH-3T3 cells. (**A**) Injury was determined on irradiated CFBE and irradiated CFBE with the addition of NIH-3T3 cells (6 × 10^5^) after wounding. Wound closure was evaluated at 0 h, 24 h and 48 h. Bar = 100 μm. (**B**) Percentage of wound size after 24 and 48 h. Data are represented as a percentage of the initial area of the wound (t0) considered as 100%. *** *p* < 0.001 CFBE + NIH-3T3 vs. CFBE at 24 and 48 h. (**C**) Wound-closure rate (μm/h) over 24 h and 48 h time intervals. *** *p* < 0.001 CFBE + NIH-3T3 vs. CFBE T. 0–24 h. Data were collected from five experiments and are shown as mean ± SD.

**Figure 5 life-12-00756-f005:**
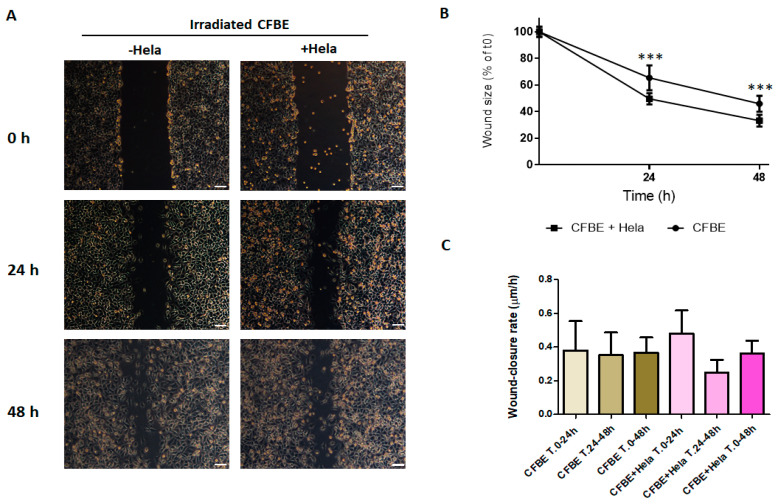
Wound closure of irradiated monolayers in the presence of HeLa cells. (**A**) Injury was determined on irradiated CFBE and irradiated CFBE with the addition of HeLa cells (6 × 10^5^) after wounding. Wound closure was evaluated at 0 h, 24 h and 48 h. Bar = 100 μm. (**B**) Percentage of wound size after 24 and 48 h. Data are represented as a percentage of the initial area of the wound (t0) considered as 100%. *** *p* < 0.001. (**C**) Wound-closure rate (μm/h) over 24 h and 48 h periods. Data were collected from five experiments and are shown as mean ± SD.

**Figure 6 life-12-00756-f006:**
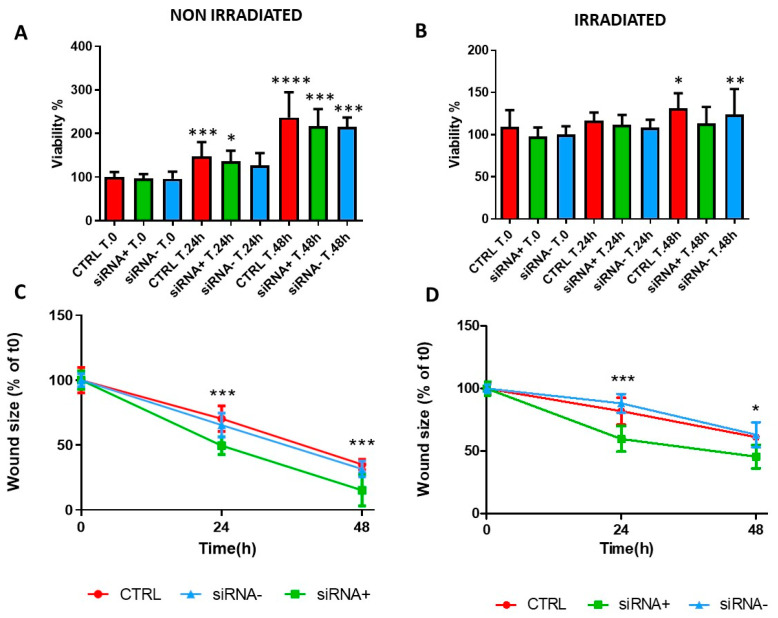
Effect of a CX43 siRNA on CFBE cell proliferation and wound closure. (**A**,**B**) CFBE cells were transfected with CX43 active siRNA (siRNA+) or scrambled siRNA (siRNA-) and evaluated for proliferation by the MTT assay at t0, t24 h and t48 h. Untransfected cells (CTRL) at time 0 were considered as 100%. Each condition (CTRL, siRNA+, siRNA−) was compared for different time points. * *p* < 0.05; ** *p* < 0.01; *** *p* < 0.001; **** *p* < 0.0001. Data were collected from two experiments and are shown as mean ± SD. (**C**,**D**) Percentage of wound size after 24 h and 48 h. Data are represented as a percentage of the initial area of untransfected cells at t0 considered as 100%. siRNA+ vs. CTRL at 24 h and 48 h: * *p* < 0.05; *** *p* < 0.001. Data were collected from two experiments and are shown as mean ± SD.

## Data Availability

The data that support the findings of this study are available from the corresponding author upon reasonable request.

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
