# Peer review of "Human Amniotic Mesenchymal Stem Cells and Fibroblasts Accelerate Wound Repair of Cystic Fibrosis Epithelium"

_life, 2022, doi:10.3390/life12050756_

Round 1
Reviewer 1 Report
The article is well written. The research design is appropriated. References must be up dated
Author Response
Reviewer: The article is well written. The research design is appropriated. References must be up dated.
Answer: Dear Reviewer, thanks for your appraisal of our article. References have been up dated in the Introduction and in the Discussion Sections.
Reviewer 2 Report
Thank you for giving me the possibility to review this manuscript.
The authors used a scratch wound assay to study wound repair in using monolayers of cystic fibrosis cells (CFBE, homozygous for the F508del mutation) in the presence of hAMSCs before the wound, or after the induction of the wound. They also -irradiated the CBFE cells to block CFBE cell proliferation. In addition, siRNA was used to downregulate CX43 expression in CFBE cells. They show that (i) CFBE cells exhibited a delayed repair as compared with wt-CFTR cells; (ii) addition of hAMSCs enhanced the wound repair rate of wounded CFBE cell monolayers, especially when added post-wounding and, (iii) CX43 down-regulation accelerated CFBE wound repair rate without affecting cell proliferation. To determine whether fibroblasts had a similar effect, the authors used NIH-3T3 fibroblasts that are CX43+ and added them to wounded cells. hAMSCs and NIH-3T3 fibroblasts, increased wound closure of irradiated CFBE monolayers.
Based on the above results, the authors concluded that hAMSCs and fibroblasts enhance the repair of a wounded CF airway epithelium, likely through a CX43-mediated mechanism mainly involving cell migration.
This is a very interesting small study. What is missing is in vivo studies using cystic fibrosis animal models and use of amniotic stem cells. Although this is a far fetched idea, it would give definitive answer on whether this kind of approach is feasible and translatable.
Author Response
Reviewer:
Thank you for giving me the possibility to review this manuscript.
The authors used a scratch wound assay to study wound repair in using monolayers of cystic fibrosis cells (CFBE, homozygous for the F508del mutation) in the presence of hAMSCs before the wound, or after the induction of the wound. They also -irradiated the CBFE cells to block CFBE cell proliferation. In addition, siRNA was used to downregulate CX43 expression in CFBE cells. They show that (i) CFBE cells exhibited a delayed repair as compared with wt-CFTR cells; (ii) addition of hAMSCs enhanced the wound repair rate of wounded CFBE cell monolayers, especially when added post-wounding and, (iii) CX43 down-regulation accelerated CFBE wound repair rate without affecting cell proliferation. To determine whether fibroblasts had a similar effect, the authors used NIH-3T3 fibroblasts that are CX43+ and added them to wounded cells. hAMSCs and NIH-3T3 fibroblasts, increased wound closure of irradiated CFBE monolayers.
Based on the above results, the authors concluded that hAMSCs and fibroblasts enhance the repair of a wounded CF airway epithelium, likely through a CX43-mediated mechanism mainly involving cell migration.
This is a very interesting small study. What is missing is in vivo studies using cystic fibrosis animal models and use of amniotic stem cells. Although this is a far fetched idea, it would give definitive answer on whether this kind of approach is feasible and translatable.
Answer: On behalf of all co-authors, we would like to thank you for the appraisal of our study. Indeed, you are right proposing the CF animal model for the evaluation of therapeutic potential of hAMSCs. This would be a defintive proof. We have now included a sentence concerning this issue in the Discussion Section, citing our previous work on a bacterial infection mouse model generating wound in the airway epithelium and homing of stem cells.